# Effect of Prophylactic Amiodarone Infusion on the Recurrence of Ventricular Arrhythmias in Out-of-Hospital Cardiac Arrest Survivors: A Propensity-Matched Analysis

**DOI:** 10.3390/jcm8020244

**Published:** 2019-02-13

**Authors:** Byung Kook Lee, Chun Song Youn, Youn-Jung Kim, Seung Mok Ryoo, Kyung Soo Lim, Gi-Byoung Nam, Su Jin Kim, Won Young Kim

**Affiliations:** 1Department of Emergency Medicine, Chonnam National University Hospital, Gwangju 61469, Korea; bbukkuk@hanmail.net; 2Department of Emergency Medicine, College of Medicine, The Catholic University of Korea, Seoul 06591, Korea; ycs1005@catholic.ac.kr; 3Department of Emergency Medicine, Ulsan University College of Medicine, Asan Medical Center, Seoul 05505, Korea; yjkim.em@gmail.com (Y.-J.K.); chrisryoo@naver.com (S.M.R.); kslim@amc.seoul.kr (K.S.L.); 4Department of Internal Medicine, Ulsan University College of Medicine, Asan Medical Center, Seoul 05505, Korea; gbnam@amc.seoul.kr; 5Department of Emergency Medicine, College of Medicine, Korea University, Seoul 02841, Korea

**Keywords:** amiodarone, cardiac arrest, targeted temperature management, ventricular arrhythmia, outcome

## Abstract

Amiodarone is recommended for shock-refractory ventricular arrhythmia during resuscitation; however, it is unknown whether amiodarone is effective for preventing ventricular arrhythmia recurrence in out-of-hospital cardiac arrest (OHCA) survivors treated with targeted temperature management (TTM). We investigated the effectiveness of prophylactic amiodarone in preventing ventricular arrhythmia recurrence in OHCA survivors. Data of consecutive adult non-traumatic OHCA survivors treated with TTM between 2010 and 2016 were extracted from prospective cardiac arrest registries of four tertiary care hospitals. The prophylactic amiodarone group was matched in a 1:1 ratio by using propensity scores. The primary outcome was ventricular arrhythmia recurrence requiring defibrillation during TTM. Among 295 patients with an initially shockable rhythm and 149 patients with initially non-shockable-turned-shockable rhythm, 124 patients (27.9%) received prophylactic amiodarone infusion. The incidence of ventricular arrhythmia recurrence was 11.26% (50/444). Multivariate analysis showed prophylactic amiodarone therapy to be the independent factor associated with ventricular arrhythmia recurrence (odds ratio 1.95, 95% CI 1.04–3.65, *p* = 0.04), however, no such association was observed (odds ratio 1.32, 95% CI 0.57–3.04, *p* = 0.51) after propensity score matching. In this propensity-score-matched study, prophylactic amiodarone infusion had no effect on preventing ventricular arrhythmia recurrence in OHCA survivors with shockable cardiac arrest. Prophylactic amiodarone administration must be considered carefully.

## 1. Introduction

Despite recent advances in critical care, the mortality rate of out-of-hospital cardiac arrest (OHCA) remains high [1]. Although the optimal target temperature and duration are unknown, targeted temperature management (TTM) at 32–36 °C for 24 h is known to reduce mortality and improve neurologic outcomes after cardiac arrest, especially, when the initial rhythm is ventricular fibrillation (VF) or ventricular tachycardia (VT) [2,3,4]. Post-cardiac arrest care should focus on optimizing cardiopulmonary function, minimizing reperfusion injury, controlling the body temperature, treating the underlying cause, and preventing re-arrest [5].

Mild hypothermia is known to have arrhythmogenic propensity mostly associated with significant prolongation of the PR and QTc intervals and the presence of J waves [6,7,8,9]. Although the incidence of VF recurrence after cardiac arrest is not well known, a large clinical trial showed that VF occurred in 8.4% of the 33 °C group and 7.4% of the 36 °C group during the first seven days [4]. However, the issue of whether to initiate or continue anti-arrhythmic therapy after the return of spontaneous circulation (ROSC) from cardiac arrest to prevent recurrent ventricular arrhythmia (VF or pulseless VT) has not yet been addressed.

Amiodarone has been shown to be effective for both the termination of shock-refractory ventricular arrhythmia during resuscitation and the prevention of sudden cardiac death after ventricular arrhythmia has subsided [5,10]. However, there is no evidence for the preventive effect of amiodarone on the recurrence of ventricular arrhythmia in OHCA patients who were successfully resuscitated and treated with TTM. Therefore, we tested the hypothesis that prophylactic amiodarone administration after ROSC could reduce the recurrence of ventricular arrhythmia in OHCA survivors.

## 2. Methods

### 2.1. Setting and Study Population

This multicenter, retrospective, observational, registry-based study was conducted at four urban emergency departments of university-affiliated teaching hospitals in Korea. Data were extracted from OHCA registries that contain prospectively collected data of consecutive patients with OHCA between January 2010 and December 2016 [11]. Patients who met the following criteria were included in the OHCA registry: non-traumatic OHCA, age ≥18 years, sustained ROSC (defined as the return of evident signs of circulation for >20 consecutive minutes), and treatment with TTM. We included patients with initial shockable rhythm and non-shockable-turned-shockable rhythm. Patients without initial rhythm records and those who had no documented shockable rhythm during cardiopulmonary resuscitation (CPR) were excluded. Prophylactic amiodarone was defined as a continuous intravenous infusion of amiodarone (1 mg/minute for 6 h, followed by 0.5 mg/minute for 18 h) after sinus rhythm conversion on ROSC for preventing ventricular arrhythmias. Throughout the study period, the use of prophylactic amiodarone therapy after ROSC was on the discretion of an emergency physician or interventional cardiologist. The institutional review board of St Mary’s hospital reviewed the study protocol and approved the study with waived informed consent (IRB No: XC13RIMI0002K).

### 2.2. TTM Protocol

In all comatose OHCA survivors, TTM was induced with intravenous cold saline and cooling devices such as Blanketrol II (Cincinnati Subzero Products, Cincinnati, OH, USA), Arctic Sun Energy Transfer Pad (Medivance Corp, Louisville, CO, USA), or an endovascular cooling device (Thermoguard; ZOLL Medical Corporation, Chelmsford, MA, USA). The target temperature of 33 °C or 36 °C was maintained for 24 h and rewarmed at a rate of 0.25 °C/h following maintained normothermia until 72 h from ROSC. The temperature was monitored with an esophageal temperature probe or rectal temperature probe. We used propofol, benzodiazepine, and opioids for sedation and analgesia. If required, a neuromuscular blocker was administered to control shivering. All patients received standard intensive care according to institutional protocols.

### 2.3. Data Collection and Outcome

Demographic and clinical data including resuscitation profiles were obtained. Laboratory values on admission were retrieved from the TTM registry. The ROSC to re-ventricular arrhythmia time was defined as the interval from ROSC to the recurrence of the first ventricular arrhythmia during TTM. The primary outcome was the recurrence of ventricular arrhythmia requiring defibrillation during TTM. The secondary outcome was survival discharge and good neurologic outcome, defined as cerebral performance category 1 or 2 at discharge.

### 2.4. Statistical Analysis

Continuous variables were expressed as mean ± standard deviation (SD) or median with the interquartile range as appropriate. Categorical variables were expressed as numbers and percentages. Student’s *t*-test was used to compare the means of normally distributed continuous variables, whereas the Mann–Whitney *U*-test was used to compare non-normally distributed continuous variables. The chi-square or Fisher’s exact test was used to compare categorical variables. A paired *t*-test or sign test were used to compare continuous variables in propensity-score-matched groups, and McNemar’s test was used for categorical variables.

To reduce the effect of treatment selection bias and potential confounding factors, we adjusted for differences in the baseline characteristics of patients by using propensity score matching. The propensity scores were estimated without regard to outcomes through a multiple logistic regression analysis. A full non-parsimonious model was developed that included all variables shown in Table 1 and Appendix A. The discrimination and calibration abilities of each propensity score model were assessed using the C-statistic and the Hosmer–Lemeshow statistic [12]. Propensity-score-matched pairs were created by matching between patients in the non-prophylactic amiodarone and prophylactic amiodarone groups on the logit of the propensity score, by using calipers of width equal to 0.2 of the standard deviation (SD) of the logit of the propensity score. By using the matched set, we examined the similarity between the non-prophylactic amiodarone group and the prophylactic amiodarone group by calculating standardized differences for each of the baseline variables listed in Table 1. In the propensity-matched analysis, the risks of recurrent shockable arrest were compared using the logistic regression model with the generalized estimating equation method, which accounted for the clustering of subjects. All tests in this study were two-sided, and a *p*-value of <0.05 was considered to indicate statistical significance. All statistical analyses were performed using SPSS for Windows version 20.0 (SPSS Inc., Chicago, IL, USA).

## 3. Results

From a total of 901 non-traumatic OHCA survivors treated with TTM, 295 patients with initially shockable OHCA and 149 patients with shockable rhythm during CPR from initially non-shockable OHCA were finally included. Of the 444 patients, 124 patients (27.9%) who received prophylactic amiodarone infusion were categorized into the prophylactic group and the other 320 patients were included in the non-prophylactic amiodarone group, see Figure 1. The median time interval from ROSC to amiodarone infusion was 113.0 [29.5–265.0] min, and the total amount of amiodarone infusion was 984.0 [885.3–1330.5] mg. The most common drugs at amiodarone infusion initiation were dopamine (81.1%), norepinephrine (60.7%), and nitrate (25.8%).

### 3.1. Comparison between the Prophylactic and Non-Prophylactic Groups

There was no difference in the demographic findings, past cardiac diseases such as previous cardiac arrest and coronary syndrome, and initial cardiac serum markers between the two groups, see Table 1. The prophylactic group showed higher rates of non-cardiac diseases such as chronic renal disease and malignancy than the non-prophylactic group. Appendix A shows the comparison of CPR profiles and post-cardiac arrest care between the prophylactic and non-prophylactic groups. Patients in the prophylactic group had more bystander CPR and longer low-flow time, whereas the usage rate of defibrillation and CPR drugs was higher in the prophylactic group than in the non-prophylactic group. The prophylactic group showed more arrhythmogenic electrocardiography patterns (ventricular premature complex, prolonged QTc interval, and non-sustained VT) and more stenotic lesions in each coronary artery than the non-prophylactic group, see Appendix A. The number of coronary artery stenosis was also higher in the prophylactic group (median, 1.0 vs. 1.5, *p* < 0.001), see Appendix A. Compared with the non-prophylactic group, patients who received prophylactic amiodarone infusion were more likely to be administered inotropic drugs during TTM.

### 3.2. Recurrence of Ventricular Tachyarrhythmia

Recurrence of ventricular arrhythmia occurred in 11.2% (33 of 295) of patients with initially shockable OHCA and 11.4% (17 of 149) of patients with shockable rhythm during CPR from initially non-shockable OHCA, see Figure 1. The median time from ROSC to the recurrence of ventricular arrhythmia was 6.3 [4.0–23.1] h. TTM induction was the most common period of ventricular arrhythmia recurrence (54.0%). The recurrence of ventricular arrhythmia was significantly higher in the prophylactic amiodarone group than in the non-prophylactic amiodarone group (16.9% vs. 9.1%, *p* = 0.02). The detailed outcome results are presented in the Appendix A. However, the rate of survival and good neurologic outcome did not show any difference, see Table 2.

The results of univariate and multivariable analyses of recurrent ventricular arrhythmia are provided in the Appendix A. A logistic regression analysis model was applied to all patients, and the crude odds ratio (OR) of prophylactic amiodarone infusion for recurrent ventricular arrhythmia was 2.046 (95% confidence interval [CI] 1.117–3.746, *p* = 0.02). After adjustment for several factors showing *p* < 0.10 in univariate analysis, prophylactic amiodarone (OR 1.946, 95% CI 1.038–3.647, *p* = 0.04) was the independent factor for the recurrence of ventricular arrhythmia requiring defibrillation during TTM.

### 3.3. Analysis of Propensity-Score-Matched Groups

Each group included 93 of 186 patients after propensity score matching for demographic data, prehospital, and in-hospital CPR variables, medications, and initial laboratory and electrocardiographic findings, See Appendix A. The recurrence of the ventricular arrest rhythm did not show any difference between the two matched groups, and the rate of survival and good neurologic outcome at discharge also did not show any significant difference, see Table 2. Moreover, prophylactic amiodarone therapy was not revealed as the independent associated factor for recurrent shockable arrest rhythm by using the logistic regression model in matched groups (OR 1.32, 95% CI 0.57–3.04, *p* = 0.51), see Table 3.

## 4. Discussion

We found that prophylactic amiodarone infusion after achieving ROSC in cardiac arrest survivors who received TTM treatment was independently associated with a higher recurrence of ventricular arrhythmias during 72 h of post-cardiac arrest care. However, the recurrence of ventricular arrhythmia was not different between the prophylactic amiodarone group and the non-prophylactic amiodarone group in a smaller matched group based on propensity scores. Survival discharge and neurologic outcome were not associated with prophylactic amiodarone irrespective of adjustment.

A recent randomized double-blind trial that compared parenteral amiodarone, lidocaine, and placebo for refractory VF/VT during cardiac arrest failed to prove the effectiveness of amiodarone or lidocaine in improving survival or neurologic outcome, but demonstrated that amiodarone improved the survival to discharge in a subgroup of patients who had bystander-witnessed cardiac arrest [13]. However, the prophylactic use of amiodarone after ROSC in cardiac arrest has not been investigated, whereas randomized trials including a large number of patients with acute myocardial infarction proved that prophylactic amiodarone reduced the incidence of VF or arrhythmic death irrespective of left ventricular dysfunction [14,15]. The use of prophylactic anti-arrhythmic medication after ROSC has limited evidence.

Pleiotropic effects, including the improvement of cardiac metabolic efficiency after the ischemic-reperfusion period were reported to have a main role in preventing reperfusion ventricular arrhythmias, whereas an increase in transmural dispersion of repolarization was reported to be associated with development of ventricular arrhythmia in animal studies [16,17,18]. Amiodarone theoretically can prolong QTc and therapeutic hypothermia can potentiate the arrhythmogenic risk of amiodarone through QTc prolongation [19]. However, an amiodarone-related adverse event is rare, and a systematic review showed that amiodarone achieved comparable rates of ROSC with placebo in severe hypothermia animal models [20,21]. In the present study, prophylactic amiodarone was independently associated with increased ventricular arrhythmias. It is postulated that worse clinical characteristics might affect the association between prophylactic amiodarone and increased ventricular arrhythmias, despite adjustments for covariates. The prophylactic amiodarone group had less bystander CPR, higher defibrillation times, higher incidence of prolonged QTc interval, more stenotic lesions in the coronary artery, higher number of coronary artery stenosis, longer low-flow time, and required more vasopressors. In other words, physicians might administer prophylactic amiodarone to patients who are expected to have recurrent ventricular arrhythmias. The independent associations of the ventricular premature complex on electrocardiography after ROSC and epinephrine use during TTM with recurrent ventricular arrhythmias in the present study also support that explanation. Coronary artery stenosis is known as a risk factor for the recurrence of cardiac events and consequently could influence the survival outcome. However, the number of coronary artery stenosis as well as the affected coronary artery were not significantly associated with the recurrent shockable arrest rhythm in our study, see Appendix A.

To reduce the effect of treatment selection bias and potential confounding factors inherent in an observational study, we compared the incidence of ventricular arrhythmias between the prophylactic amiodarone group and the non-prophylactic amiodarone group in a propensity-score-matched cohort and finally observed that prophylactic amiodarone had no association with ventricular arrhythmias during 72 h after ROSC. The incidences of recurrent ventricular arrhythmias were 11.3% and 13.4% in the total cohort and in the propensity-score-matched cohort in the present study, respectively. A Hypothermia after Cardiac Arrest Group study showed a 34% incidence of lethal or long-lasting arrhythmia, and a TTM trial showed a 17% incidence of VT and 8% incidence of VF [2,3]. The insignificant association between prophylactic amiodarone and ventricular arrhythmias in the present study might be attributed to the lower incidence of ventricular arrhythmias. The majority of the time during 72 h after ROSC was spent for TTM. Thereby, prophylactic amiodarone was mostly infused to hypothermic patients. The action of prophylactic amiodarone might be influenced by lower body temperature, as amiodarone had a comparable anti-arrhythmic effect to placebo in a hypothermic animal model [20].

OHCA patients treated with amiodarone during CPR required more vasopressor or treatment for bradycardia than those who were treated with a placebo [22]. Another randomized trial also showed that patients with refractory VF/VT treated with amiodarone required more temporary cardiac pacing; however, other amiodarone-related adverse events were not different [13]. The prophylactic amiodarone group required more dopamine and norepinephrine in the present study. It is assumed that the patients in the prophylactic group may have been in a more severe condition; however, amiodarone-related adverse effects such as bradycardia or hypotension may cause worse clinical characteristics in the prophylactic amiodarone group.

A previous study demonstrated that prophylactic lidocaine was associated with less recurrent arrhythmic arrest and was eventually found to be independently associated with a higher survival discharge [23]. However, survival discharge and neurologic outcome were not associated with prophylactic amiodarone in the present study. The prophylactic lidocaine group had lower recurrent arrhythmic arrest and higher survival discharge in the total cohort of the previous study [23]. That might lead to a misinterpretation that lower recurrent arrhythmic arrest after prophylactic lidocaine is responsible for a higher survival discharge. However, the prophylactic group had comparable survival discharge despite showing lower recurrent arrhythmic arrest in the propensity-matched cohort [23]. Clinical outcomes such as survival discharge and neurologic outcome seem to not be singly dependent on the use of prophylactic anti-arrhythmic drugs.

This study has several limitations. Owing to the retrospective nature of the analysis, we report an independent association, but not a causal relationship, between prophylactic amiodarone use and recurrent ventricular arrhythmias. As the attending physician decided the use of prophylactic amiodarone, patients who required greater administrations of defibrillation during CPR tended to receive prophylactic amiodarone. Thereby, the prophylactic amiodarone group had worse clinical characteristics, which led to a selection bias. OHCA patients with shockable rhythm and non-shockable-turned-shockable rhythm were analyzed together in the present study. Initial non-shockable rhythm is considered an ominous factor, and late-occurring shockable rhythm can result in a delay in ROSC, which might contribute to poor outcomes. The crude OR of prophylactic amiodarone for recurrent ventricular arrhythmias in patients with initial shockable rhythm was significant, whereas that in patient with non-shockable-turned-shockable rhythm was not significant in the present study. However, a randomized trial including shockable rhythm only and another randomized trial including non-shockable-turned-shockable rhythm only reported a consistent result that amiodarone during CPR tended to achieve a higher survival to discharge, although the difference was insignificant [13,24].

## 5. Conclusions

Prophylactic amiodarone after successful resuscitation from cardiac arrest with initial shockable or subsequently occurring shockable rhythm was not associated with the prevention of recurrent ventricular tachyarrhythmias during TTM in our propensity-score matched cohort. Routine use of prophylactic amiodarone administration may be nonbeneficial for preventing the recurrence of ventricular arrhythmia in OHCA survivors. Further randomized clinical trials would be warranted to define guidelines for recommending the initiation of prophylactic amiodarone therapy for cardiac arrest with shockable rhythm.

## Figures and Tables

**Figure 1 jcm-08-00244-f001:**
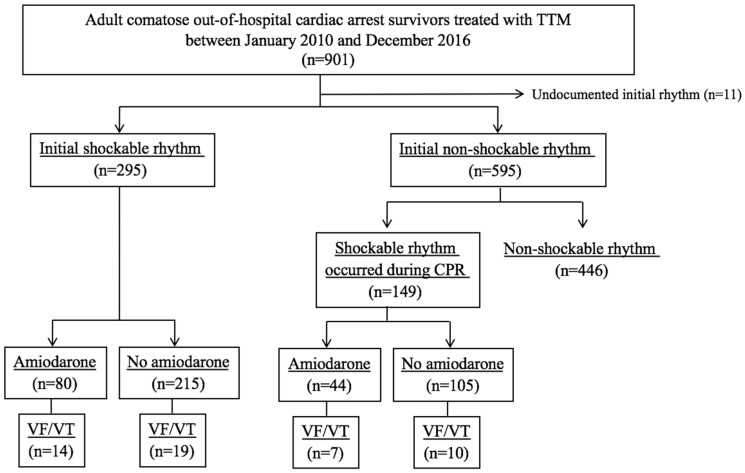
Flowchart of patient selection. TTM, targeted temperature management; CPR, cardiopulmonary resuscitation; VF, ventricular fibrillation; VT, ventricular tachycardia.

**Table 1 jcm-08-00244-t001:** Baseline and clinical characteristics and laboratory findings.

Variables	Total (*N* = 444)	No Prophylactic Amiodarone (*n* = 320)	Prophylactic Amiodarone (*n* = 124)	*p* Value
Age, years	55.0 [45.0–65.0]	56.0 [46.0–65.0]	53.5 [42.3–64.0]	0.20
Male	333 (75.0)	242 (75.6)	91 (73.4)	0.63
Past medical history				
History of cardiac arrest	7 (1.6)	6 (1.9)	1 (0.8)	0.68
Acute coronary syndrome	79 (17.8)	59 (18.4)	20 (16.1)	0.57
Arrhythmia	27 (6.1)	16 (5.0)	11 (8.9)	0.13
Hypertension	160 (36.0)	128 (40.0)	32 (25.8)	0.005
Diabetes	88 (19.8)	64 (20.0)	24 (19.4)	0.88
Chronic pulmonary disease	10 (2.3)	5 (4.0)	5 (1.6)	0.15
Chronic renal disease	18 (4.1)	17 (5.3)	1 (0.8)	0.03
Liver cirrhosis	3 (0.7)	3 (0.9)	0 (0.0)	0.56
Malignancy	14 (3.2)	14 (4.4)	0 (0.0)	0.009
Vital signs				
Systolic pressure, mmHg	119.5 [93.8-–141.0]	116.0 [90.0–142.0]	120.0 [100.0–140.0]	0.79
Diastolic pressure, mmHg	70.5 [60.0–90.0]	71.0 [60.0–90.0]	70.0 [60.0–90.0]	0.58
Pulse rate, beats/min	101.3 ± 27.7	101.4 ± 27.3	101.0 ± 29.9	0.89
Body temperature, °C	36.1 [35.5–36.4]	36.1 [35.5–36.4]	36.0 [35.3–36.4]	0.26
Laboratory findings, initial				
White blood cell, × 103/μL	13.4 [10.6–18.1]	13.3 [10.7–18.0]	13.7 [10.5–18.6]	0.68
Hemoglobin, g/dL	14.2 [12.4–15.4]	13.9 [12.1–15.3]	14.6 [12.8–15.7]	0.03
Sodium, mmol/L	141.0 [138.0–143.0]	141.0 [138.0–143.0]	141.0 [138.0–143.0]	0.62
Potassium, mmol/L	3.8 [3.4–4.4]	3.9 [3.4–4.4]	3.6 [3.3–4.3]	0.02
Calcium, mg/dL	8.0 [7.4–8.7]	8.0 [7.3–8.7]	8.1 [7.3–8.8]	0.11
Magnesium, mg/dL	2.2 [2.0–2.6]	2.2 [1.9–2.5]	2.4 [2.1–2.7]	0.004
Troponin-I, ng/mL	0.605 [0.116–4.800]	0.650 [0.124–4.870]	0.529 [0.101–3.460]	0.49
CK-MB, ng/mL	7.9 [2.8–30.4]	7.8 [2.9–25.1]	8.8 [2.3–43.8]	0.58
BNP, pg/mL	129.6 [41.0–691.0]	117.0 [40.0–448.2]	183.0 [42.0–1357.5]	0.17

Values are expressed as means ± standard deviation, medians [interquartile range], or numbers (%). Abbreviations: CK-MB, creatinine kinase MB fraction; BNP, B-natriuretic peptide.

**Table 2 jcm-08-00244-t002:** Outcome characteristics.

Outcome	Total Data (*N* = 444)			Matched Data (*n* = 186)		
	No Prophylactic Amiodarone (*n* = 320)	Prophylactic Amiodarone (*n* = 124)	*p*-Value	No Prophylactic Amiodarone (*n* = 93)	Prophylactic Amiodarone (*n* = 93)	*p*-Value
Recurrent shockable arrest	29 (9.1)	21 (16.9)	0.02	11 (11.8)	14 (15.1)	0.51
Survival discharge	236 (73.8)	98 (79.0)	0.25	72 (77.4)	73 (78.5)	0.87
Good neurologic outcome	155 (48.4)	71 (57.3)	0.10	48 (51.6)	55 (59.1)	0.26

Values are expressed as number (%).

**Table 3 jcm-08-00244-t003:** Logistic regression analysis of prophylactic amiodarone infusion for recurrent shockable arrest rhythm.

Model	*N*	Odds Ratio	95% Confidence Interval	*p*-Value
Crude	444	2.046	1.117–3.746	0.02
Multivariate adjusted	441	1.946	1.038–3.647	0.04
Propensity score matching	186	1.321	0.574–3.043	0.51

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
