# Peer review of "Effect of Prophylactic Amiodarone Infusion on the Recurrence of Ventricular Arrhythmias in Out-of-Hospital Cardiac Arrest Survivors: A Propensity-Matched Analysis"

_jcm, 2019, doi:10.3390/jcm8020244_

Reviewer 1 Report

The main aim of this study is evaluation of effectiveness of prophylactic amiodarone administration for preventing ventricular arrhythmia recurrence in out-of-hospital cardiac arrest (OHCA) survivors treated with targeted temperature management (TTM). The results of this study are interesting and intriguing and may potentially have important implications for management of OHCA survivors that are treated with TTM. The manuscript is well-written, and the data partly support the authors' conclusions. However, addressing the following comments will greatly aid the reader in the interpretation and implications of the findings of the study.

Comments

1.    There seem to be a number of differences in the cardiopulmonary resuscitation and clinical profiles between the two groups (amiodarone vs. no amiodarone), including the use of medications at the time of CPR, the low-flow time, initial ECG abnormalities, the extent of coronary artery disease (CAD), and the use of inotropic drugs. Many if not all these factors could influence the risk of arrhythmias in OHCA survivors with or without amiodarone use. Therefore, though the authors have accounted for the potential differences in the baseline and clinical characteristics and laboratory findings (Table 1) between the groups, important determinants of recurrent arrhythmias such as the extent of CAD, and the LVEF have not been included in the analysis, fading the author’s conclusions. In fact, as the authors note, the amiodarone group seemed to have a more severe cardiac condition, than the control group. I would recommend the authors to

(1) add ejection fraction in their model of analysis;

(2) include an additional data and analysis on the median number of affected coronary vessels in the groups;

(3) add data on and analysis based on the etiology of cardiac arrest.

2.    Please add the causes of in-hospital death. How many of the recurrent ventricular arrhythmias in each of the groups had a fatal outcome?

3.    Please revise or comment on this sentence: ‘Pleiotropic effects, including improvement of cardiac metabolic efficiency after the ischemic-reperfusion period and increase in transmural dispersion of repolarization, were reported to have a main role in preventing reperfusion ventricular arrhythmias in animal studies [16-18].’ The ‘increase in transmural dispersion of repolarization’ is quite unclear since an increase in transmural dispersion of repolarization can manifests in the electrocardiogram as an ST segment elevation (or idiopathic J wave) and even lead to ventricular arrhythmias, including of the Torsade de Pointes type. I would like the authors to clarify what they mean with this sentence.

4.    Please revise your conclusions to reflect the hypothesis. The conclusions in the abstract are more suitable for the hypothesis and results of the present study.

Author Response

Response to Reviewer 1 Comments

The main aim of this study is evaluation of effectiveness of prophylactic amiodarone administration for preventing ventricular arrhythmia recurrence in out-of-hospital cardiac arrest (OHCA) survivors treated with targeted temperature management (TTM). The results of this study are interesting and intriguing and may potentially have important implications for management of OHCA survivors that are treated with TTM. The manuscript is well-written, and the data partly support the authors' conclusions. However, addressing the following comments will greatly aid the reader in the interpretation and implications of the findings of the study.

Comments

1.    There seem to be a number of differences in the cardiopulmonary resuscitation and clinical profiles between the two groups (amiodarone vs. no amiodarone), including the use of medications at the time of CPR, the low-flow time, initial ECG abnormalities, the extent of coronary artery disease (CAD), and the use of inotropic drugs. Many if not all these factors could influence the risk of arrhythmias in OHCA survivors with or without amiodarone use. Therefore, though the authors have accounted for the potential differences in the baseline and clinical characteristics and laboratory findings (Table 1) between the groups, important determinants of recurrent arrhythmias such as the extent of CAD, and the LVEF have not been included in the analysis, fading the author’s conclusions. In fact, as the authors note, the amiodarone group seemed to have a more severe cardiac condition, than the control group. I would recommend the authors to

(1) add ejection fraction in their model of analysis;

Response 1-(1):

Thank you for your comment. Per your suggestion, we have added data in Table S1 as below:

Table S1. Cardiopulmonary resuscitation profiles and post-cardiac arrest care between the non-prophylactic amiodarone and prophylactic amiodarone groups

Variables

No prophylactic amiodarone

(n=320)

Prophylactic amiodarone

(n=124)

P Value

Left ventricular ejection fraction (%) during TTM, n=217

n=157, 55.0 [40.0-62.0]

n=60, 52.2 [36.3-59.9]

0.39

Values are expressed as medians [interquartile range].

(2) include an additional data and analysis on the median number of affected coronary vessels in the groups;

Response 1-(2):

Thank you for your comment. Per your suggestion, we have added data in Table S1 as below:

Table S1. Cardiopulmonary resuscitation profiles and post-cardiac arrest care between the non-prophylactic amiodarone and prophylactic amiodarone groups

Variables

No prophylactic amiodarone

(n=320)

Prophylactic amiodarone

(n=124)

P Value

Coronary artery angiography

210 (65.6)

90 (72.6)

0.16

  Left anterior descending   stenosis

78 (37.1)

59 (65.6)

< 0.001

  Right coronary artery stenosis

67 (31.9)

47 (52.2)

0.001

  Left circumflex artery   stenosis

61 (29.0)

45 (50.0)

0.001

Significant coronary stenosis

<0.001< span="">

    1 vessel

56 (26.7)

20 (22.2)

    2 vessels

18 (8.6)

4 (4.4)

    3 vessels

38 (18.1)

41 (45.6)

  Number of coronary vessels

1.0 [0.0-2.0]

1.5 [0.0-3.0]

<0.001< span="">

Values are expressed as medians [interquartile range], or numbers (%).

(3) add data on and analysis based on the etiology of cardiac arrest.

Response 1-(3):

Thank you for your comment. Per your suggestion, we have added data in Table S1 as below:

Table S1. Cardiopulmonary resuscitation profiles and post-cardiac arrest care between the non-prophylactic amiodarone and prophylactic amiodarone groups

Variables

No prophylactic amiodarone

(n=320)

Prophylactic amiodarone

(n=124)

P Value

Etiology of cardiac   arrest

0.90

Presumed cardiac cause

279 (87.2)

110 (88.7)

Respiratory cause

8 (2.5)

3 (2.4)

Other medical condition

33 (10.3)

11 (8.9)

Values are expressed as number (%).

Also, we have analyzed outcome data based on etiology of cardiac arrest as below:

Table A. Outcome characteristics based on etiology of cardiac arrest

Etiology of cardiac arrest

No prophylactic amiodarone

(n=320)

Prophylactic amiodarone

(n=124)

P Value

Presumed cardiac cause

n = 279 (87.2)

n = 110 (88.7)

Recurrent shockable arrest

27 (9.7)

17 (15.5)

0.11

Survival discharge

221 (79.2)

91 (82.7)

0.43

Good neurologic outcome

149 (53.4)

68 (61.8)

0.13

Respiratory cause

n = 8 (2.5)

n = 3 (2.4)

Recurrent shockable arrest

0 (0)

1 (33.3)

0.27

Survival discharge

2 (25.0)

1 (33.3)

>0.99

Good neurologic outcome

0 (0)

0 (0)

Other medical condition

n = 33 (10.3)

n = 11 (8.9)

Recurrent shockable arrest

2 (6.1)

3 (27.3)

0.09

Survival discharge

13 (39.4)

6 (54.5)

0.49

Good neurologic outcome

6 (18.2)

3 (27.3)

0.67

Values are expressed as number (%).

2.    Please add the causes of in-hospital death. How many of the recurrent ventricular arrhythmias in each of the groups had a fatal outcome?

Response2:

Thank you for your comment. We have analyzed the cause of in-hospital death as below (Table S2). Cardiovascular and sepsis were main causes of in-hospital death in our study patients.

Table S2. Detailed outcome results

Variables

No prophylactic amiodarone

(n=320)

Prophylactic amiodarone

(n=124)

P Value

Causes of in-hospital death

n = 84 (26.3)

n = 26 (21.0)

0.17

  Cardiovascular

28 (33.3)

13 (50.0)

  Cerebral

8 (9.5)

1 (3.8)

  Sepsis

28 (33.3)

10 (38.5)

  Others or undetermined

20 (23.8)

2 (7.7)

 Values are expressed as number (%).

We have analyzed outcome data of the patients with recurrent shockable arrhythmia as below (Table S2). The rate of return of spontaneous circulation in the prophylactic and the non-prophylactic amiodarone group did not show any difference (93.1% vs. 100%, P=0.50). Two patients in non-prophylactic amiodarone group with recurrent shockable arrhythmia (2/29, 6.9%) died without return of spontaneous circulation.

Table S2. Detailed outcome results

Variables

No prophylactic amiodarone

(n=320)

Prophylactic amiodarone

(n=124)

P Value

Recurrent shockable arrest

n = 29 (9.3)

n = 21 (16.9)

Outcome of recurrent shockable arrhythmia

ROSC

27 (93.1)

21 (100)

0.50

No ROSC

2 (6.9)

0

Survival discharge

18 (62.1)

17 (81.0)

0.15

In-hospital death

11 (37.9)

4 (19.0)

Values are expressed as number (%).

Abbreviations: ROSC, return of spontaneous circulation.

3.    Please revise or comment on this sentence: ‘Pleiotropic effects, including improvement of cardiac metabolic efficiency after the ischemic-reperfusion period and increase in transmural dispersion of repolarization, were reported to have a main role in preventing reperfusion ventricular arrhythmias in animal studies [16-18].’ The ‘increase in transmural dispersion of repolarization’ is quite unclear since an increase in transmural dispersion of repolarization can manifests in the electrocardiogram as an ST segment elevation (or idiopathic J wave) and even lead to ventricular arrhythmias, including of the Torsade de Pointes type. I would like the authors to clarify what they mean with this sentence.

Response3:

We appreciate your comment. We have revised our sentence as below:

“Pleiotropic effects, including improvement of cardiac metabolic efficiency after the ischemic-reperfusion period were reported to have a main role in preventing reperfusion ventricular arrhythmias, whereas increase in transmural dispersion of repolarization was reported to be associated with development of ventricular arrhythmia in animal studies [16-18].”

4.    Please revise your conclusions to reflect the hypothesis. The conclusions in the abstract are more suitable for the hypothesis and results of the present study.

Response4:

Thank you for your comment. We revised our conclusions as below per your suggestion.

“Prophylactic amiodarone after successful resuscitation from cardiac arrest with initial shockable or subsequently occurring shockable rhythm was not associated with the prevention of recurrent ventricular tachyarrhythmias during TTM in our propensity-score matched cohort. Routine use of prophylactic amiodarone administration may be nonbeneficial for preventing the recurrence of ventricular arrhythmia in OHCA survivors. Further randomized clinical trials would be warranted to define guidelines for recommending the initiation of prophylactic amiodarone therapy for cardiac arrest with shockable rhythm.”

Reviewer 2 Report

I have no comment. The manuscript is well design and made. The only limitation is its retrospective nature but given the addressed issue I believe that it could be unethical to procede with a RCT

Author Response

We appreciate your comment.

Reviewer 3 Report

Lee et al. examined the effect of Amiodarone on ventricular arrhythmia recurrence in out-of-hospital cardiac arrest survivors that received targeted temperature management (TTM). Patients were categorized into shockable (n=295) or non-shockable with subsequent shockable during CPR (n=149). Of these 444 patients, approximately 124 received Amiodarone and 320 did not receive amiodarone. While the rate of shockable reccurrence was higher in the Amiodarone group compared to non-amiodarone, survival discharge and neurological outcome were not statistically different. Moreover, no difference in ventricular arrhythmia were noted between groups. This manuscript was relatively straightforward and the experimental design appears appropriate for the questions asked. Outside of minor spell checks, I have no comments regarding this manuscript. 

Author Response

We appreciate your comment. We have corrected the typos in our manuscript.

Round  2

Reviewer 1 Report

The authors tried to provide additional comparison between the groups in response to my  comments regarding their original submission; however, their efforts do not fully answer the questions I raised. 

1.      Have the authors included the LVEF, CAD vessels affected, etc. in their statistical model? No such data is provided. Simple comparison between the groups does not show whether they will influence the outcome or not.

2.      The highly significant difference in CAD complexity between the groups, which has been revealed by this additional analysis, is clinically important since it can influence the outcome and the recurrence of cardiac events. Please comment on this in the results section and in the discussion (Table S1).

Author Response

Response to Reviewer 1 Comments

The authors tried to provide additional comparison between the groups in response to my comments regarding their original submission; however, their efforts do not fully answer the questions I raised. 

1.      Have the authors included the LVEF, CAD vessels affected, etc. in their statistical model? No such data is provided. Simple comparison between the groups does not show whether they will influence the outcome or not.

Response 1: Thank you for your comment. "CAD vessels affected" had been already in the propensity matching analysis. According to your valuable comment, we reviewed LVEF but it did not significant difference between two groups. We did not include LVEF and did not matching again.We have added the results of number of coronary artery stenosis and LVEF in Table S3 as below.

Table S3. Logistic regression analysis for recurrent shockable arrest rhythm

 Variables

 Univariate analysis

Multivariable analysis

Odds

ratio

95% CI 

P-value

Odds

ratio

95% CI

P-value

Lower

Upper

Lower

Upper

Coronary artery angiography

  Number of significant coronary artery stenosis

0.820

0.598

1.123

0.22

Left ventricular ejection fraction (%) during TTM, n=217

1.008

0.982

1.034

0.56

2.      The highly significant difference in CAD complexity between the groups, which has been revealed by this additional analysis, is clinically important since it can influence the outcome and the recurrence of cardiac events. Please comment on this in the results section and in the discussion (Table S1).

Response 2:Thank you for your comment. We have added the result in the results section as below.

“The number of coronary artery stenosis was also higher in the prophylactic group (median, 1.0 vs. 1.5, P<0.001) (Table S1).”

We have also added the sentences in the discussion section as below.

“The prophylactic amiodarone group had less bystander CPR, higher times of defibrillation, higher incidence of prolonged QTc interval, more stenotic lesions in the coronary artery, higher number of coronary artery stenosis, and longer low flow time, and required more vasopressors.”

“Coronary artery stenosis is known as a risk factor for the recurrence of cardiac events and consequently could influence the survival outcome. However, the number of coronary artery stenosis as well as the affected coronary artery were not significantly associated with the recurrent shockable arrest rhythm in our study (Table S3).”